# Lung Ultrasound Patterns in Multisystem Inflammatory Syndrome in Children (MIS-C)-Characteristics and Prognostic Value

**DOI:** 10.3390/children9070931

**Published:** 2022-06-21

**Authors:** Anna Camporesi, Marco Gemma, Danilo Buonsenso, Stefania Ferrario, Anna Mandelli, Matteo Pessina, Veronica Diotto, Elena Rota, Irene Raso, Laura Fiori, Alessandro Campari, Francesca Izzo

**Affiliations:** 1Department of Pediatric Anesthesia and Intensive Care, Children’s Hospital “Vittore Buzzi”, 20154 Milano, Italy; stefania.ferrario@asst-fbf-sacco.it (S.F.); anna.mandelli64@asst-fbf-sacco.it (A.M.); matteo.pessina@asst-fbf-sacco.it (M.P.); veronica.diotto@asst-fbf-sacco.it (V.D.); elena.rota@asst-fbf-sacco.it (E.R.); francesca.izzo@asst-fbf-sacco.it (F.I.); 2Department of NeuroAnesthesia and NeuroIntensive Care, Fondazione IRCCS Istituto Neurologico Carlo Besta, 20154 Milano, Italy; marco.gemma@istituto-besta.it; 3Department of Woman and Child Health and Public Health, Fondazione Policlinico Universitario “A. Gemelli”, 00168 Roma, Italy; danilo.buonsenso@policlinicogemelli.it; 4Department of Pediatric Cardiology, Children’s Hospital “Vittore Buzzi”, 20154 Milano, Italy; irene.raso@asst-fbf-sacco.it; 5Department of Pediatrics, Children’s Hospital “Vittore Buzzi”, 20154 Milano, Italy; laura.fiori@asst-fbf-sacco.it; 6Department of Pediatric Radiology, Children’s Hospital “Vittore Buzzi”, 20154 Milano, Italy; alessandro.campari@asst-fbf-sacco.it

**Keywords:** multisystem inflammatory syndrome in children, lung ultrasound, COVID-19

## Abstract

Objective and design: Following COVID-19 infection, children can develop an hyperinflammatory state termed Multisystem Inflammatory Syndrome in Children (MIS-C). Lung Ultrasound (LUS) features of COVID-19 in children have been described, but data describing the LUS findings of MIS-C are limited. The aim of this retrospective observational study conducted between 1 March and 31 December 2020, at a tertiary pediatric hospital in Milano, is to describe LUS patterns in patients with MIS-C and to verify correlation with illness severity. The secondary objective is to evaluate concordance of LUS with Chest X-ray (CXR). Methodology: Clinical and laboratory data were collected for all patients (age 0–18 years) admitted with MIS-C, as well as LUS and CXR patterns at admission. PICU admission, needed for respiratory support and inotrope administration, hospital, and PICU length of stay, were considered as outcomes and evaluated in the different LUS patterns. An agreement between LUS and CXR evaluation was assessed with Cohen’ k. Results: 24 children, who had a LUS examination upon admission, were enrolled. LUS pattern of subpleural consolidations < or > 1 cm with or without pleural effusion were associated with worse Left Ventricular Ejection Fraction at admission and need for inotropes. Subpleural consolidations < 1 cm were also associated with PICU length of stay. Agreement of CXR with LUS for consolidations and effusion was slight. Conclusion: LUS pattern of subpleural consolidations and consolidations with or without pleural effusion are predictors of disease severity; under this aspect, LUS can be used at admission to stratify risk of severe disease.

## 1. Introduction

Although considered initially relatively spared from Coronavirus-19 disease (COVID-19), children can develop a life-threatening hyperinflammatory state 4–6 weeks after primary infection. This novel entity has been termed Multisystem Inflammatory Syndrome in Children (MIS-C) and comprises fever, gastrointestinal symptoms, evidence of mucocutaneous inflammation, and a high level of inflammatory markers in blood. A subset of patients develop cardiac involvement, including myocarditis, myocardial dysfunction, and coronary artery changes [1,2].

The respiratory involvement in MIS-C is very early and frequent, with tachypnea, desaturation, and increase of the respiratory effort as the main symptoms [3]; respiratory support (non-invasive or invasive) can be necessary in some cases [2].

The use of bedside lung ultrasonography (LUS) has increased worldwide for the assessment of patients presenting with acute respiratory failure [4].

This tool is also increasingly used in pediatrics, being non-invasive, repeatable, and safe because of lack of radiation exposure. Moreover, it has very good sensitivity and specificity for common respiratory patterns in pediatric population [5,6], and it has been recently employed as a clinical tool to detect cardiopulmonary interactions in acutely ill children with systemic diseases [7].

While the LUS features of COVID-19 in children have been described [8,9,10,11], data describing the LUS findings of MIS-C are limited [12].

The aim of the study is to characterize the LUS patterns in patients with a diagnosis of MIS-C and to verify if LUS findings could predict illness severity and need of support.

The secondary objective is to describe concordance of LUS with Chest X-ray (CXR) in the cohort.

## 2. Materials and Methods

This is a retrospective observational study conducted at a tertiary pediatric hospital in Milano, Italy, between the 1st of March and the 31st of December 2020. Patients younger than 18 years with a diagnosis of MIS-C according to CDC guidelines [1] and requiring hospitalization were enrolled. The study was approved by the Ethic Committee of our Institution (2021/ST/005). Written informed consent was obtained by patients’ caregivers.

For every patient we recorded demographic data, symptoms at admission, duration of fever before admission and global duration of fever, Left Ventricular Ejection Fraction (LVEF, %) obtained with Simpson’s method [13] at admission and worst LVEF value, lung ultrasound patterns upon admission, Chest X-ray evidence of consolidation and/or effusion, laboratory tests (N-terminal pro b-type natriuretic peptide (NT-pro-BNP, pg/mL), C Reactive Protein (CRP, mg/L), Troponin T (TNT, ng/L), Procalcitonin (PCT, ng/mL) serum ferritin (mcg/L), and D-Dimer (mcg/L) at admission and their worst values during stay.

Additionally, admission to pediatric intensive care unit (PICU), Pediatric Index of Mortality 2 [14] (PIM2) days of PICU stay, need for respiratory or cardiovascular support, and relative duration were noted.

## 3. Lung and Cardiac Ultrasound Examination

Lung ultrasound examinations have been performed by 3 intensivists, who have received standard training in LUS (Winfocus PNCUS BL1P) and have at least 3 years of experience with LUS technique. When an intensivist’s evaluation was required for MIS-C patients, the scans were performed upon admission to the Emergency Department as part of the clinical-instrumental examination. Exams were carried out with a high frequency (12–3 MHz) linear probe (Affiniti 70, Philips, Amsterdam, The Netherlands) at the patient’s bedside, using a modified three-zone per hemithorax Bedside Lung Ultrasound in Emergency protocol (described by Lichtenstein [4]) (Figure 1). Each hemithorax was divided into anterior, lateral, and posterior zones, and upper and lower zones (divided by the inter-nipple line); anterior aspect of the chest was identified by the anterior axillary line, lateral aspect of the chest by the anterior and posterior axillary lines, and posterior aspect of the chest by the posterior axillary line and the spine, not including the scapular area.

Children were scanned in a recumbent or semirecumbent position and (when clinically feasible) rolled onto their side or placed in a sitting position to optimize posterior scanning.

In order to minimize the spread of the virus and other microbes, LUS was performed with the probe protected by a single-use plastic cover, and an ultrasound transmission gel in a single-use package. After the examination, all the material was cleaned correctly.

We used the following definitions to characterize the LUS patterns [15]:▪A-lines: normal appearance of horizontal, equidistant, parallel artefacts originating at regular intervals from the pleural line (visceral and parietal pleura)▪B-lines: laser-like signals arising from the hyper-echoic pleural line, extending to the bottom of the screen without fading and moving synchronously with respiration▪Subpleural consolidation: Small, triangular, or oval shaped, echo-poor region adjacent to the pleura without a tissue-like pattern (AB)▪Lobar consolidation: Large, hypoechogenic region adjacent to the pleura with a tissue-like pattern (AB) and an irregular pleural border▪Pleural effusion: Anechoic or hypoechoic collection external to lung parenchyma, typically in a dependent lung region with or without respiratory movement of the lung within the effusion (flapping lung)

Using the mentioned LUS semeiotics, lung aeration and lung pattern were classified following a 5 point score:(1)Normal lung sliding, regular pleural line, and A lines(2)Vertical artifacts, pleural line indented with several B-lines per field in the posterior regions(3)Vertical artifacts, pleural line indented with several B-lines per field in all regions(4)Broken pleural line with subpleural consolidations < 1 cm(5)Consolidations > 1 cm with or without pleural effusion (Table 1).

Cardiac ultrasound upon admission and during the hospitalization was performed by the cardiologist on call in the Emergency Department or in PICU immediately after admission.

### Statistical Analysis

Data were analyzed with the R software version 4.1.0 (18 May 2021)—“Camp Pontanezen”

Copyright (C) 2021 The R Foundation for Statistical Computing Platform: R Core Team (2021). R: A language and environment for statistical computing. R Foundation for Statistical Computing, Vienna, Austria. URL https://www.R-project.org/ (accessed on 18 May 2021).

The occurrence of PICU admission, the need for respiratory support, and the need for inotrope administration were considered as the dependent variables in three respective logistic regression models. Univariate variable selection (Likelihood Ratio test) using *p* < 0.25 to select candidates for the multivariable model was performed on the following possible predictive variables: age (yrs), male gender, BMI, LVEF, NT-pro-BNP, CRP, TNT, PCT, serum ferritin, and D-Dimer on admission; the time between fever appearance and admission and the global duration of fever; a LUS pattern 3, 4, 5, or 4 and/or 5 on admission (no/yes variables); CXR evidence of consolidation or of effusion (no/yes variables).

Similarly, two multivariate linear regression models were built, taking the EF on admission and the worst EF registered during the hospital stay as the dependent variables and a multivariate Poisson regression model taking the number of days of hospital stay as the dependent variable was built. A similar analysis was undertaken on the patients admitted to the PICU, taking the number of days of PICU stay as the dependent variable. The goodness-of-fit chi-squared test on the residual difference is reported for Poisson models.

The agreement between LUS and CXR evaluation was assessed with Cohen’ k (unweighted, since our analysis dealt with dichotomous variables only). The agreements between CXR diagnosis of “consolidation” and LUS 4, 5, and 4 and/or 5 were studied, as well as the agreement between between CXR diagnosis of “effusion” and LUS 4, 5, and 4 and/or 5.

Continuous data are reported as mean ± SD (median (IQR)). Categorical data are reported as number (percentage). Odds ratios and incidence rate ratios are reported as OR (95% CI) and IRR ((95% CI), respectively.

## 4. Results

### 4.1. Demographic

Thirty-eight children (27M/11 F) were admitted to our institution from March to December 2020 with a diagnosis of MIS-C. Twenty-five (65.8%) were admitted to our PICU. Twenty-four received LUS upon admission to ED and were enrolled in the study.

After the Caucasian one, Hispanic ethnicity (28%) was the most frequent one; 12/38 children presented with comorbidities (prematurity, full term baby but small for gestational age, iron-deficient anaemia, congenital CMV, mental delay, idiopathic juvenile arthritis); in 3 of these cases the comorbidity was clinical obesity.

Fever (100%) and abdominal symptoms (97%) were the most common at presentation; 52% of the patients also presented respiratory symptoms (mostly tachypnea and desaturation, <92% in room air). Cardiac Function was normal (LVEF > 55%) at admission in 17/38 patients (45%); 9/38 had LVEF between 45% and 54%, 8/38 (21%) had LVEF between 35–44%; 5/38 (13%) presented with LVEF < 34%.

All patients had IgG anti-SARS-CoV-2 and 8 patients had positive nasopharyngeal swab for COVID-19. In no patient was the primary COVID-19 infection severely symptomatic and nobody had been hospitalized due to that. In about half the cases, the patients were unaware of their previous COVID-19 infection, and in the remaining, the primary infection had been paucisymptomatic.

Twenty-four patients received LUS examination upon admission; of these, 22 were admitted to PICU and 2 were admitted to inpatient ward. Twenty-four patients also received CXR on the same day as the LUS examination.

Two patients presented a LUS pattern type 1, 14 a LUS pattern type 2, 6 showed a LUS pattern type 3, 12 showed a LUS pattern type 4, and 6 showed a LUS pattern type 5.

Some patients showed more than one simultaneous pattern.

Table 2 summarizes the clinical characteristics of the cohort according to LUS patterns and the necessary support. Figure 2 shows the laboratory exams in the different LUS groups. Figure 3 shows LVEF upon admission and worst LVEF during hospital stay according to LUS patterns.

The multivariate logistic models showed that BMI (OR 2.89(1.38–12.67) and EF on admission (OR 0.77(0.52–0.92), *p* = 0.042) were significantly associated with PICU admission, while age (OR 1.37(1.07–1.93), *p* = 0.032) and D-Dimer on admission (OR 1.01(1.00–1.02), *p* = 0.039) were significantly associated with need of respiratory support.

LUS pattern of 4 and/or 5 on admission (coeff. −12.57, *p* = 0.008) and the global duration of fever (1-day coeff. 2.31, *p* = 0.036) (Adjusted R-squared 0.39, whole model *p* = 0.003) were significantly associated with worse LVEF on admission. Conversely, concerning the ability to predict the worst LVEF during the hospital stay, multivariate linear regression model showed that age (coeff. −0.54, *p* = 0.010), the LVEF on admission (1% coeff. 0.76. *p* < 0.001), and the global duration of fever (1-day coeff. 1.27, *p* = 0.033) (Adjusted R-squared 0.82, whole model *p* < 0.001) were significant predictors.

The multivariate Poisson regression model found that BNP (IRR 1.01(1.00–1.02), *p* = 0.008) and the occurrence of consolidation on CXR upon admission (IRR 1.30(1.04–1.61), *p* = 0.018) were significant predictors of the whole length of admission (goodness-of-fit chi-squared test on the residual OR 10.00(1.66–89.80), *p* = 0.020) difference *p* = 0.998), while in the subanalyses of the 25 children needing PICU, D-Dimer on admission (IRR 1.01(1.00–1.02), *p* = 0.010) and a LUS of 4 (IRR 1.69(1.06–2.74), *p* = 0.030) were associated with length of PICU admission (goodness-of-fit chi-squared test on the residual difference *p* = 0.950).

LUS of 4 and/or 5 on admission was significantly associated with the need inotrope administration with the lowest AIC, while we did not find variables that significantly predicted the duration of mechanical ventilation.

### 4.2. Agreement with CXR

The agreement of CXR consolidation diagnosis with LUS 4 (Cohen’s k = 25) and with LUS 4 and/or 5 (Cohen’s k = 24) was fair, while it was slight with LUS 5 (Cohen’s k = 12).

The agreement of CXR effusion diagnosis with LUS 4 (Cohen’s k = 0.08), LUS 5 (Cohen’s k = 0.05), and LUS 4 and/or 5 (Cohen’s k = 14) was slight.

Table 3 describes a correlation between LUS and CXR findings.

## 5. Discussion

In this study, we found that specific clinical, laboratory, and ultrasonographic parameters are significant predictors of clinically-relevant outcomes in children with MIS-C. Although LUS is an increasingly used tool to predict outcomes in the ICU [16], this is, to our knowledge, the first report of the use of LUS as a predictor of outcome in patients presenting with MIS-C.

Only two patients in our cohort presented with a pattern 1; they were not admitted to PICU. This is of interest because it could support our hypothesis of correlation between severity of LUS features and need of care.

Interestingly, the finding of subpleural and lobar consolidations with or without pleural effusion was associated with a lower LVEF on admission and need for inotrope use; also, presence of a LUS pattern of subpleural consolidations was associated with PICU length of stay.

These specific patterns of LUS that were associated with the aforementioned outcomes are not the ones we would expect in patients who present with acute heart failure, such as those with MIS-C. Lung ultrasound has been proven useful in acute heart failure depicting pulmonary congestion, that is, accumulation of extravascular lung water (EVLW) (a term used to describe water within the lungs but outside pulmonary vasculature) [17]. This accumulation of EVLW occurs due to fluid retention in patients with reduced ejection fraction and/or fluid redistribution to the lungs in patients with preserved ejection fraction and a noncompliant cardiovascular system [18].

B-lines are the typical lung ultrasound pattern encountered in patients with acute heart failure [19] and represent the pre-alveolar stage of cardiogenic pulmonary edema, when secondary interlobular septa are congested but there is still no alveolar flooding [20]. Assessment of EVLW by B-lines provides an excellent alternative to chest X-ray, and correlates with the invasive methods of evaluation of EVLW as wedge pressure measurement [17,20,21,22].

It would be therefore expected to find LUS patterns in 2 or 3 (that is, B lines in different degrees of severity) of these patients who presented with acute heart failure in the context of MIS-C. However, the lung ultrasound pattern we encountered more resembled a feature of Acute Respiratory Distress Syndrome (ARDS) with inhomogeneous sonographic interstitial syndrome (SIS), which has been defined [23] by the presence of multiple focal, patched, or diffuse vertical artifacts (B-lines) fanning out from the lung–wall interface and gravitational consolidations.

Subpleural consolidations, on the other side, as well as lobar consolidations, have been described in viral pneumonitis and bronchiolitis, pneumonia, acute respiratory distress syndrome [24], and pulmonary infarction [25], but are not described in cardiogenic pulmonary edema [18].

The ability to distinguish infectious consolidations of bacterial origin from viral consolidation, noninfectious consolidation, and atelectasis can be extremely challenging, and in these patients, all these conditions could be present at the same time.

In many studies [25,26], consolidations >1 cm with air or fluid bronchograms are considered diagnostic for bacterial CAP (community acquired pneumonia), but it is still debated whether a small lung consolidation can be found with viral etiology [26]. Of course, other typical signs of viral bronchiolitis or pneumonia are the presence of a thick irregular pleura, the presence of confluent B lines, and the absence of air bronchogram.

Subpleural consolidations can also be present in a diffuse microembolic/microthrombotic event. This phenomenon is common in COVID-19 patients [27] because of thrombotic occlusion of small- to mid-sized pulmonary arteries with subsequent infarction of lung parenchyma and can show at LUS a triangular hypoechoic consolidation with sharp margins in the absence of air bronchograms [28].

Subpleural and lobar consolidations could also be due to the advanced process of ARDS in which the transition from vertical artifacts to consolidation is a continuum with something similar to the transition between the ground glass and consolidation in chest CT, where ground glass is due to thickening of the interstitium and/or the presence of fluid and/or the presence of collapsed areas and/or increased circulation [29].

Our patients showed the presence of subpleural consolidations or consolidations with or without pleural effusion correlating with the degree of heart failure (LVEF at admission and need for inotropes), and this is a new finding. Subpleural consolidations are common in COVID-19 pneumonia in children [8] and have also been described in asymptomatic pediatric patients [30,31]. It is possible that this pattern in MIS-C patients refers to sequelae of the primary infection that had not been detected during initial SARS-CoV-2 infection but was still present at time of MIS-C diagnosis. It is also possible that they are expressions of the inflammatory state that these patients show through fever and elevated markers of inflammation [32]. The correlation of subpleural consolidations with PICU length of stay could also be interpreted under this light as a sign of a more severe disease. MIS-C is a novel entity that follows the primary infection with COVID-19 in children which is still under investigation under different aspects. The inflammatory state in this disease seems caused from a cytokine storm that causes fever, elevation of all markers of inflammation and multiorgan failure, with activation of T cells, macrophages, natural killer cells and overproduction of immune or non-immune defense cells, and the release inflammatory cytokines and chemical mediators. Endothelial cell dysfunction follows, resulting in damage to the microvascular system and abnormal activation of the coagulation system, resulting in systemic small vessel vasculitis and extensive microthrombosis [33]. Therefore, it is possible to speculate that the pattern we found on LUS is a possible expression of a mix of pathological events on the lung parenchyma, a consequence of both the cardiogenic edema and a direct inflammatory/micro-embolic lung disease.

Additionally, the presence of lung consolidations in our series could be interpreted as an effect of impaired lung aeration in the context of a globally “wet” lung, or as expression of overinfection (atelectasis vs pneumonia), which is, even with LUS, sometimes very difficult to distinguish between [34,35].

We acknowledge the scale we used to classify LUS pattern in patients with MIS-C is not an ordinal but rather a nominal one, and under this regard it is interesting to note that some classes of patients, namely LUS 3 and 5, show worst LVEF, highest inotrope use, highest percentage of ventilation, longest hospital or PICU LOS, and ventilation days, although not reaching statistical significance in the multivariate analysis. MIS-C is a multifactorial disease and overall severity can be due to different factors, not only lung involvement. In this regard, it is possible that a more complex score, which includes multiple clinical, laboratory, and ultrasound factors, may better provide a prognostic role. Additionally, although there is increasing knowledge of the physical bases of LUS artifacts [36,37] and that LUS can provide a semi-quantitative measure of lung aeration [38,39], there is not yet international consensus on the most appropriate scores to assign to each LUS pattern. In any case, it is noteworthy that patients within some specific LUS patterns (those 3 to 5) are those with a more severe MIS-C profile, and therefore our study provides preliminary data for the design of future, larger, prospective studies on the role of LUS in MIS-C patients.

It is reported that elevation of ferritin is of negative prognostic value in patients with MIS-C as an index of hyper-inflammation state [40]. In our series ferritin levels are higher in LUS pattern 5, as if the presence of consolidations and/or effusion was due to the inflammatory vasculitic process with the production of pleuric exudative fluid.

Concerning the other factors which have found association with outcomes in our study, BMI has already been identified as a risk factor for intensive care unit admission in adult patients with COVID-19 [41,42].

D-Dimer has already been underlined in MIS-C, not only as a typical laboratory result [31] in these patients, but also as a predictor of severity being higher in severe cases of MIS-C [43,44] and correlating with PICU admission and oxygen supplementation in another study [45]. Our results confirm these data, D-Dimer in our series being a predictor of need for respiratory support and length of PICU stay.

The age of our patients was also associated with respiratory support, with older patients being more prone to need it. This might be interpreted within a well-established spectrum of more severe SARS-CoV-2 disease with increasing age [44].

Another interesting finding of our study is the poor correlation between Chest X ray and LUS in diagnosing consolidations and pleural effusion. Several studies have already highlighted how LUS has better sensitivity in children than conventional diagnostic for pneumonia [34] as well as for small consolidations in bronchiolitis [46]. Our results are in line with literature under this aspect. It is possible that CXR, obtained upon admission together with LUS examination, was too early to show what LUS can detect sooner. Interestingly, and also in the series presented by Hameed [12], around half the cases had normal CXR at admission, which then progressed.

Later on, during the hospital stay, our patients were not been examined with CXR anymore as this has been judged useless given their clinical improvement and the biological cost, so that there is no possible comparison with another CXR available.

In fact, there is evidence in available literature that LUS artifacts are generated already after microscopic alternations in the porosity of the peripheral lung [36], being therefore more sensitive and able to detect abnormalities at an earlier stage compared with CXR.

We can confirm, as described by other authors [12], that CXR findings in these patients include peribronchial cuffing and perihilar interstitial thickening until perihilar bilateral consolidation with pleural effusion. These findings are typical of pulmonary edema secondary to cardiac dysfunction, but also to fluid overload and hypoalbuminemia, or a combination thereof. Interestingly, comparing with ultrasound, the most frequent images in our series were pattern 2 (posterior B lines) and pattern 4 (subpleural consolidations), instead of pattern 3 (diffuse B lines in all fields) as expected.

No comparison with Computed Tomography (CT) chest imaging is possible in our series, as it was deemed not necessary to scan our patients.

## 6. Limitations and Strengths

We acknowledge that this study has some limitations, the most important of which is its retrospective nature and small sample size. However, LUS was performed by experienced physicians with a standard LUS training and was always combined with a proper physical examination and other laboratory analysis. The same 3 experienced intensivists performed the LUS examinations in order to reduce the operator dependency. We believe that LUS could be really the difference that allows physicians to identify the patients who are at a higher risk of respiratory or circulatory failure evolution depending on the LUS alteration, so they could be monitored more intensively and rapidly enhance the treatment if needed.

## 7. Conclusions

MIS-C is a rather novel entity whose characteristics and physiopathology are still a matter of research. With our paper we provide an extensive description of LUS patterns in a well characterized cohort of MIS-C patients and show that the presence of subpleural consolidations and consolidations with or without pleural effusion correlates with LVEF on admission and need for inotrope use as well as PICU length of stay, and thus are useful as potential predictors of disease severity and need for intensive care treatment. Lung ultrasound is confirmed to be a safe, non-invasive, and more sensitive tool than radiography in the diagnosis of lung pathology and with its use patients are exposed to lesser degrees of X-ray, which does have a biological cost.

Our observations offer clinicians a new point-of-care clinical tool that, in addition to traditional clinical examination and laboratory tests, can support in the prediction of more severe disease and the consequent timely implementation and organization of the possibly needed resources.

## Figures and Tables

**Figure 1 children-09-00931-f001:**
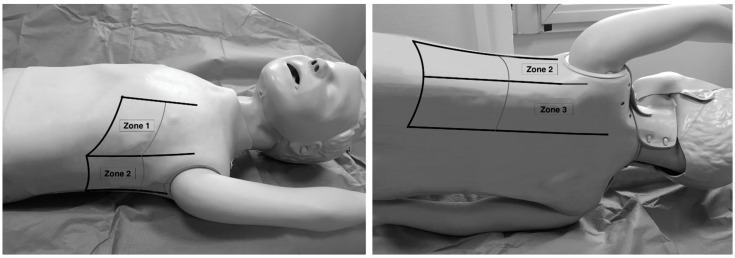
Lung ultrasound scanning protocol. Zone 1 is defined by the anterior chest wall between the anterior axillary line and sternum, zone 2 is defined by the lateral chest wall between the anterior and posterior axillary lines, and zone 3 is defined by the posterolateral chest wall between the posterior axillary line and the spine. Upper and lower zones are defined by the inter-nipple line.

**Figure 2 children-09-00931-f002:**
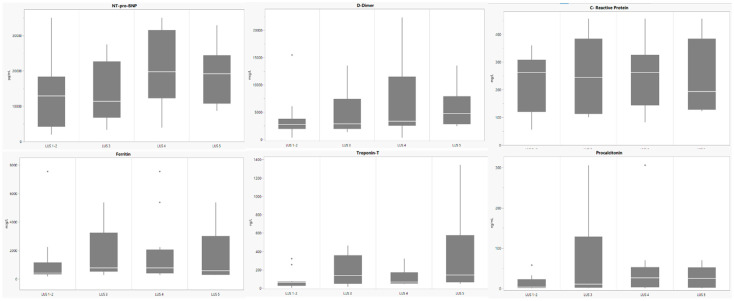
Laboratory values upon admission according to Lung Ultrasound (LUS) Groups. Dots represent outliers.

**Figure 3 children-09-00931-f003:**
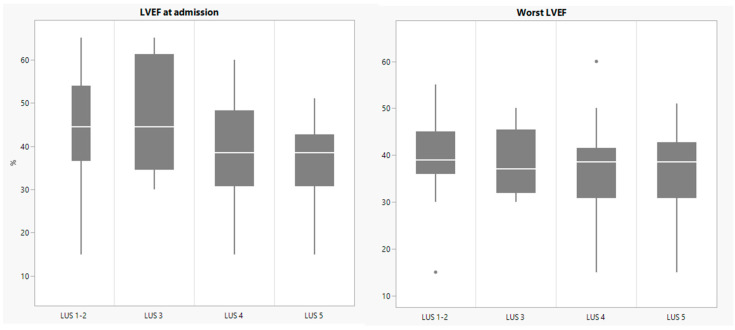
Left Ventricular Ejection Fraction (LVEF) upon admission and worst LVEF during stay, according to Lung Ultrasound (LUS) groups. Dots represent outliers.

**Table 1 children-09-00931-t001:** Classification of the ultrasound patterns.

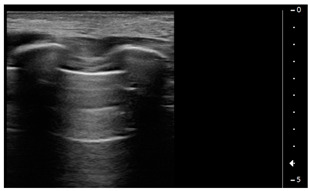	PATTERN 1 Normal lung sliding, regular pleural line andA lines
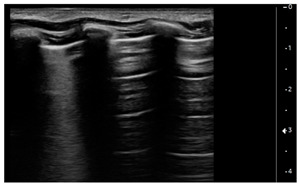	PATTERN 2 Vertical artifacts, pleural line indented with several B-lines per field in the posterior regions
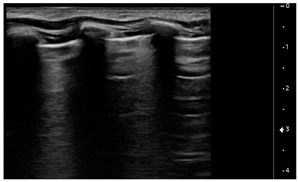	PATTERN 3 Vertical artifacts, pleural line indented with several B-lines per field in all regions
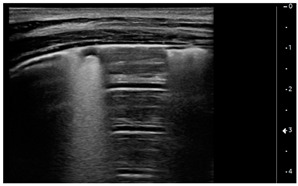	PATTERN 4 Broken pleural line with subpleural consolidations <1 cm
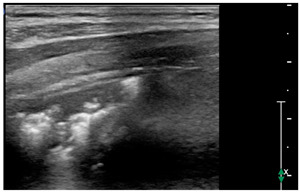	PATTERN 5 Consolidation >1 cm without pleural effusion
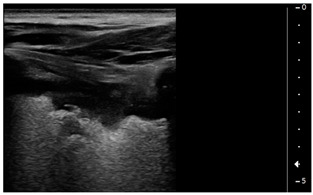	PATTERN 5 Consolidation >1 cm with pleural effusion

**Table 2 children-09-00931-t002:** Demographic, clinic, and laboratory characteristics of the cohort. BMI: Body Mass Index. Data are presented as mean ± SD (median (IQR)).

	All	LUS < 3	LUS = 3	LUS = 4	LUS = 5
Sex, M	71.1%	66.7%	83.3%	71.1%	83.3%
Age, years	8.105 ± 4.865 [8(4–12)]	7.571 ± 5.144 [8(3–12)]	11.5 ± 4.764 [12.5(10.5–13.75)]	8.105 ± 4.865 [8(4–12)]	9.333 ± 4.179 [9.5(6.25–12.75)]
Comorbidities, N(%)	12 (50%)	6 (25%)	1 (4%)	4 (16%)	2 (8%)
BMI, kg/m^2^	18.751 ± 3.546 [18.445(16.408–20.293)]	17.47 ± 3.037 [16.55(16.315–18.175)]	20.39 ± 5.216 [20.2(18.75–21.63)]	18.751 ± 3.546 [18.445(16.408–20.293)]	22.213 ± 5.078 [20.57(19.365–24.24)]
Fever before therapy, days	5.5 ± 1.765 [5(5–6)]	5.526 ± 2.195 [5(4–7)]	5.333 ± 0.816 [5.5(5–6)]	5.5 ± 1.765 [5(5–6)]	5.833 ± 1.835 [5(5–6.5)]
Fever global duration, days	7.028 ± 2.145 [7(5–9)]	7.6 ± 2.326 [7(6.75–9)]	6.5 ± 1.049 [6.5(6–7)]	7.028 ± 2.145 [7(5–9)]	7.2 ± 1.924 [7(6–8)]
PICU (yes)	65.8%	38.1%	100%	65.8%	100%
LVEF at admission, %	50.842 ± 13.219 [51.5(39.25–60)]	57.286 ± 9.624 [60(52–62)]	46.667 ± 14.137 [44.5(36.5–57.75)]	50.842 ± 13.219 [51.5(39.25–60)]	36.5 ± 11.777 [38.5(36.5–39.75)]
LVEF worst value, %	46.843 ± 12.781 [49(38.5–55)]	53.889 ± 10.922 [53.5(49–60)]	38.417 ± 7.446 [37(33.375–42.5)]	46.843 ± 12.781 [49(38.5–55)]	36.5 ± 11.777 [38.5(36.5–39.75)]
Inotrope use	41.9%	14.3%	50%	41.9%	66.7%
Ventilation (y)	39.3%	18.2%	66.7%	39.3%	50%
Hospital stay, days	12.081 ± 3.854 [12(10–14)]	10.9 ± 3.905 [10(8.5–12.25)]	14.167 ± 3.251 [14(12–16.75)]	12.081 ± 3.854 [12(10–14)]	13.5 ± 3.834 [13.5(11.25–16.5)]
PICU LOS, days	2.789 ± 2.801 [2.5(0–4)]	1.048 ± 1.564 [0(0–2)]	4.333 ± 2.066 [4.5(2.5–5.75)]	2.789 ± 2.801 [2.5(0–4)]	4.5 ± 1.378 [4(4–4.75)]
Ventilation days	2.6 ± 1.43 [2.5(1.25–3.75)]	2 ± 1.414 [2(1.5–2.5)]	3 ± 1.826 [3(1.75–4.25)]	2.6 ± 1.43 [2.5(1.25–3.75)]	2.667 ± 2.082 [2(1.5–3.5)]

**Table 3 children-09-00931-t003:** Number of patients presenting specific LUS patterns and consolidations or effusions on Chest X Ray.

	LUS 1	LUS 2	LUS 3	LUS 4	LUS 5
CXR_consolidations	0	6	6	8	4
CXR_effusions	0	5	2	4	2

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
