# Peer review of "Lung Ultrasound Patterns in Multisystem Inflammatory Syndrome in Children (MIS-C)-Characteristics and Prognostic Value"

_children, 2022, doi:10.3390/children9070931_

Round 1
Reviewer 1 Report
Introduction
Line 59-63: rewrite –correct the space between words…the sentence is to long. In general avoid using ;(semicolon) in the middle of sentence
Materials and methods
Did you also have information on the severity of the clinical presentation of COVID in these patients?
Was there a significant correlation between the severity of the COVID presentation and later occurrences of MIS-C
RESULTS
Line 186 – align text
DISCUSSION
Line 243 – 244: ..“ this is, 243 to our knowledge, the first report of the use of LUS as a predictor of outcome in patients 244 presenting with MIS-C. M“ - this is repeated many times
Line 259-260: „It would be therefore expected to find LUS pattern n 2 or 3 – B lines in different de- 259 grees of severity – in these patients who presented with acute heart failure in the context 260 of MIS-C“ ..sounds confusing
Line 262 lack of abbreviation for „Acute Respiratory Distress Syndrome“
Line 275-277: missing reference
Line 345-346: „Our results are in line 345 with literature under this aspect.“ – add those references
LIMITATIONS
In your study LUS was performed by three experienced intensivists so this is not a limitation of your study moreover it is a study strength
CONCLUSION
Sentences are also too long…you did not mention advantage of LUS use in comparison with lung X ray, especially in children
REFERENCE
equalize the font size of the references
„DOI numbers (Digital Object Identifier) are not mandatory but highly encouraged“ – put little more effort to add DOI
need to add new reference: Skopljanac I, Ivelja MP, Barcot O, Brdar I, Dolic K, Polasek O, Radic M. Role of Lung Ultrasound in Predicting Clinical Severity and Fatality in COVID-19 Pneumonia. J Pers Med. 2021 Jul 30;11(8):757. doi: 10.3390/jpm11080757.
Reviewer 2 Report
In 3. Results, 3.1.1. Demographic, please describe all comorbid conditions.
Comorbidities should be included in Table 2.
The discussion needs English editing.
Discussion Lines 339-341 need pediatric references (“More severe SARS-CoV-2 disease with increasing age”).
For a broader audience appeal reach, references 9, 10,11, and 12 are only commented on in lines 64 and 65, then on lines 346 -353, other references are cited (34, 44, 45); this discussion area needs to be rewritten and should comment at length references 9, 10, 11 and especially reference 12, Hameed S Radiology 2021; 298:e1-e10, a wider audience would appreciate if you demonstrate the comparison of other lung imaging techniques in your series of cases.
Round 2
Reviewer 1 Report
I am satisfied with the corrections made.